# Upregulation of OASIS/CREB3L1 in podocytes contributes to the disturbance of kidney homeostasis

Yoshiaki Miyake[1,10], Masanori Obana [1,2,3,4,10 ✉], Ayaha Yamamoto[1,10], Shunsuke Noda[1], Koki Tanaka[1], Hibiki Sakai[1], Narihito Tatsumoto[5], Chihiro Makino[5], Soshi Kanemoto[6], Go Shioi[7], Shota Tanaka[1], Makiko Maeda[8,9], Yoshiaki Okada[1], Kazunori Imaizumi[6], Katsuhiko Asanuma[5] & Yasushi Fujio[1,3]

Podocyte injury is involved in the onset and progression of various kidney diseases. We previously demonstrated that the transcription factor, old astrocyte specifically induced substance (OASIS) in myofibroblasts, contributes to kidney fibrosis, as a novel role of OASIS in the kidneys. Importantly, we found that OASIS is also expressed in podocytes; however, the pathophysiological significance of OASIS in podocytes remains unknown. Upon lipopolysaccharide (LPS) treatment, there is an increase in OASIS in murine podocytes. Enhanced serum creatinine levels and tubular injury, but not albuminuria and podocyte injury, are attenuated upon podocyte-restricted OASIS knockout in LPS-treated mice, as well as diabetic mice. The protective effects of podocyte-specific OASIS deficiency on tubular injury are mediated by protein kinase C iota (PRKCI/PKCι), which is negatively regulated by OASIS in podocytes. Furthermore, podocyte-restricted OASIS transgenic mice show tubular injury and tubulointerstitial fibrosis, with severe albuminuria and podocyte degeneration. Finally, there is an increase in OASIS-positive podocytes in the glomeruli of patients with minimal change nephrotic syndrome and diabetic nephropathy. Taken together, OASIS in podocytes contributes to podocyte and/or tubular injury, in part through decreased PRKCI. The induction of OASIS in podocytes is a critical event for the disturbance of kidney homeostasis.

[1] Laboratory of Clinical Science and Biomedicine, Graduate School of Pharmaceutical Sciences, Osaka University, Osaka, Japan. [2] Radioisotope Research Center, Institute for Radiation Sciences, Osaka University, Osaka, Japan. [3] Integrated Frontier Research for Medical Science Division, Institute for Open and Transdisciplinary Research Initiatives (OTRI), Osaka University, Osaka, Japan. [4] Global Center for Medical Engineering and Informatics (MEI), Osaka University, Osaka, Japan. [5] Department of Nephrology, Graduate School of Medicine, Chiba University, Chiba, Japan. [6] Department of Biochemistry, Institute of Biomedical & Health Sciences, Hiroshima University, Hiroshima, Japan. [7] Laboratory for Animal Resources and Genetic Engineering, RIKEN Center for Biosystems Dynamics Research, Kobe, Japan. [8] Laboratory of Clinical Pharmacology and Therapeutics, Graduate School of Pharmaceutical Sciences, Osaka University, Osaka, Japan. [9] Medical Center for Translational Research, Department of Medical Innovation, Osaka University Hospital, Osaka, Japan. [10]These authors contributed equally: Yoshiaki Miyake, Masanori Obana, Ayaha Yamamoto. ✉email: obana@phs.osaka-u.ac.jp

Current therapeutic approaches to kidney diseases, including acute kidney injury and chronic kidney disease, are merely supportive. To develop novel therapeutic strategies, it is important to unravel the pathological mechanisms of kidney diseases. Podocytes are highly differentiated cells that play a central role in the ultrafiltration system, together with endothelial cells and the glomerular basement membrane[1,2]. Podocytes form foot processes, which are interconnected by a slit diaphragm. Effacement of podocyte foot processes is closely associated with various proteinuric kidney diseases, such as diabetic nephropathy (DN) and nephrotic syndrome[3,4]. In addition to the cell autonomous roles of podocytes, much attention has recently been focused on the interplay between podocytes and tubular epithelial cells in the pathogenesis of kidney diseases[5–7]. Although podocytes are thought to be a therapeutic target, the molecular mechanisms underlying podocyte dysfunction still remain unclear.

Old astrocyte specifically induced substance (OASIS)/cAMP responsive element-binding protein 3-like 1 (CREB3L1) is a transcription factor of the CREB/activating transcription factor (ATF) family, whose N-terminal domain is released as an active form by various stimuli, such as endoplasmic reticulum stress[8,9], transforming growth factor beta 1 (TGF-β1)[10], and so on. Under physiological conditions, OASIS contributes to bone formation, as observed upon analysis of conventional OASIS knockout mice[11,12]. In pathological models, OASIS has been reported to play a protective role in the large intestinal mucosa, during colitis inflammation[13]. Moreover, OASIS functions as a tumor suppressor[14–16]. Recently, we provided the first evidence of the pathological significance of OASIS in the kidneys, that is, OASIS in myofibroblasts facilitates kidney fibrosis, concomitant with an increase in cell proliferation and migration[17]. Furthermore, we confirmed that OASIS was upregulated in myofibroblasts, in both human and murine fibrotic kidneys. Moreover, OASIS expression was also observed in murine glomeruli. In addition, there was an increase in *OASIS/CREB3L1* mRNA in the kidneys of focal segmental glomerulosclerosis, a nephrotic syndrome, as analyzed using the Nephroseq database (https://www.nephroseq.org/resource/login.html). Accumulating evidence suggested that transcription factors play key roles in the maintenance of podocyte homeostasis[18–20]. Therefore, in the present study, we focused on the contribution of OASIS to podocyte-associated diseases.

Here, we explored the functional roles of OASIS in podocytes using podocyte-restricted OASIS knockout and transgenic mice. The present study revealed that upregulation of OASIS in podocytes contributes to podocyte and/or tubular injury, in part through decreased protein kinase C iota (PRKCI/PKCι) expression. This is the first demonstration of the pathophysiological significance of OASIS in podocytes.

## Results

**OASIS expression increased upon lipopolysaccharide (LPS) treatment in podocytes.** To clarify the cellular regions expressing OASIS in murine glomeruli, we performed immunohistochemistry on serial sections of murine kidneys using an anti-OASIS or anti-Wilms' tumor 1 (WT-1) antibody (Fig. 1a). The results showed that the WT-1-positive cells expressed OASIS. Moreover, to ensure that OASIS is expressed in podocytes, we generated podocyte-restricted OASIS conditional knockout (OASIS cKO) mice using the *podocin* (*Nphs2*) promoter-driven Cre-loxP recombination system (Fig. 1b–d). Glomeruli were isolated from OASIS cKO and control mice, following which the transcript expression of *Oasis/Creb3l1* was evaluated using quantitative PCR. *Oasis/Creb3l1* mRNA expression decreased to less than 10% in the glomeruli of OASIS cKO mice, as compared to those of the control mice. The expression of OASIS

protein also reduced significantly in the glomeruli of the OASIS cKO mice, indicating that OASIS is expressed in murine podocytes. Next, to examine the relevance of OASIS in podocytopathy, a murine podocyte cell line was treated with LPS, which triggers podocyte injury[21–24]. Interestingly, LPS (especially at lower dose) induced/activated OASIS by 5–10-fold, with its peak levels observed at 6 h (Fig. 1e, f). We also found that the N-terminal domain of OASIS increased in the nuclear fraction of LPS-treated podocytes (Supplementary Fig. 1). Importantly, a toll-like receptor (TLR) 4 inhibitor, TAK-242[25] suppressed LPS-induced OASIS expression (Fig. 1g). Moreover, laser capture microdissection (LCM) followed by immunoblotting revealed that OASIS expression was also upregulated in the glomeruli of LPS-injected mice (Fig. 1h). The number of OASIS-positive podocytes increased after LPS treatment (Fig. 1i, j). Taken together, OASIS upregulation was associated with LPS-induced podocytopathy.

**Acute kidney dysfunction was ameliorated in the OASIS cKO mice, after LPS treatment.** To determine the pathological significance of OASIS in podocytes, the OASIS cKO and control mice were injected with LPS. Twenty-four hours after LPS treatment, their urinary albumin-creatinine ratio (ACR) and serum creatinine (sCr) levels were measured (Fig. 2). Under physiological conditions, there was no difference in ACR and sCr levels between the OASIS cKO and control mice. However, unexpectedly, in the LPS-treated group, OASIS deficiency failed to affect the ACR levels. On the other hand, the sCr level decreased significantly in the OASIS cKO mice, as compared to that in the control mice, after LPS treatment (control, $1.01 \pm 0.27$ mg/dL; cKO, $0.76 \pm 0.16$ mg/dL, $P < 0.05$), suggesting that OASIS in podocytes contributes to LPS-induced acute kidney dysfunction.

**Deletion of podocyte OASIS attenuated LPS-induced tubular injury.** Immunohistochemistry using an anti-WT-1 antibody was performed to understand the pathology in the OASIS cKO mice. The results showed that there was no difference in the number of WT-1-positive podocytes between the OASIS cKO and control mice (Fig. 3a, b). Moreover, quantitative PCR demonstrated that the transcript expression of podocyte markers, including *Nphs1* and *Nphs2*, was not changed upon deficiency of OASIS in podocytes (Fig. 3c, d). Additionally, electron microscopic analysis also revealed that the effacement of the LPS-induced podocyte foot process was unaltered in the OASIS cKO mice, relative to that in the control mice (Fig. 3e, f). When we focused on tubular epithelial cells, Periodic acid–Sciff (PAS) staining showed tubular damage; notably, tubular vacuolization was observed in the kidneys of control mice, after LPS treatment, whereas OASIS deletion in podocytes suppressed the same (Fig. 3g, h). No difference was observed in the lotus tetragonolobus lectin (LTL)-positive area between the OASIS cKO and control mice, after LPS treatment (Fig. 3i, j). In addition, *Lcn2* mRNA, a marker of tubular injury, was reduced in the OASIS cKO mice (Fig. 3k). These data indicated that OASIS in podocytes was involved in tubular damage, rather than podocyte injury, in the LPS model.

**Kidney function was preserved in the OASIS cKO mice, upon generation of the streptozotocin (STZ)-induced DN model.** Accumulating evidence demonstrated that podocytes play important roles in the pathogenesis of DN[26,27]. Moreover, it is reported that TLR4 is associated with the pathological process of DN[28]. In fact, we confirmed that the *Tlr4* mRNA level was upregulated in the kidneys of the STZ-induced DN model (Supplementary Fig. 2). Thus, we examined the effects of podocyte OASIS in the kidneys of the STZ-induced DN model. LCM and immunoblotting showed that the

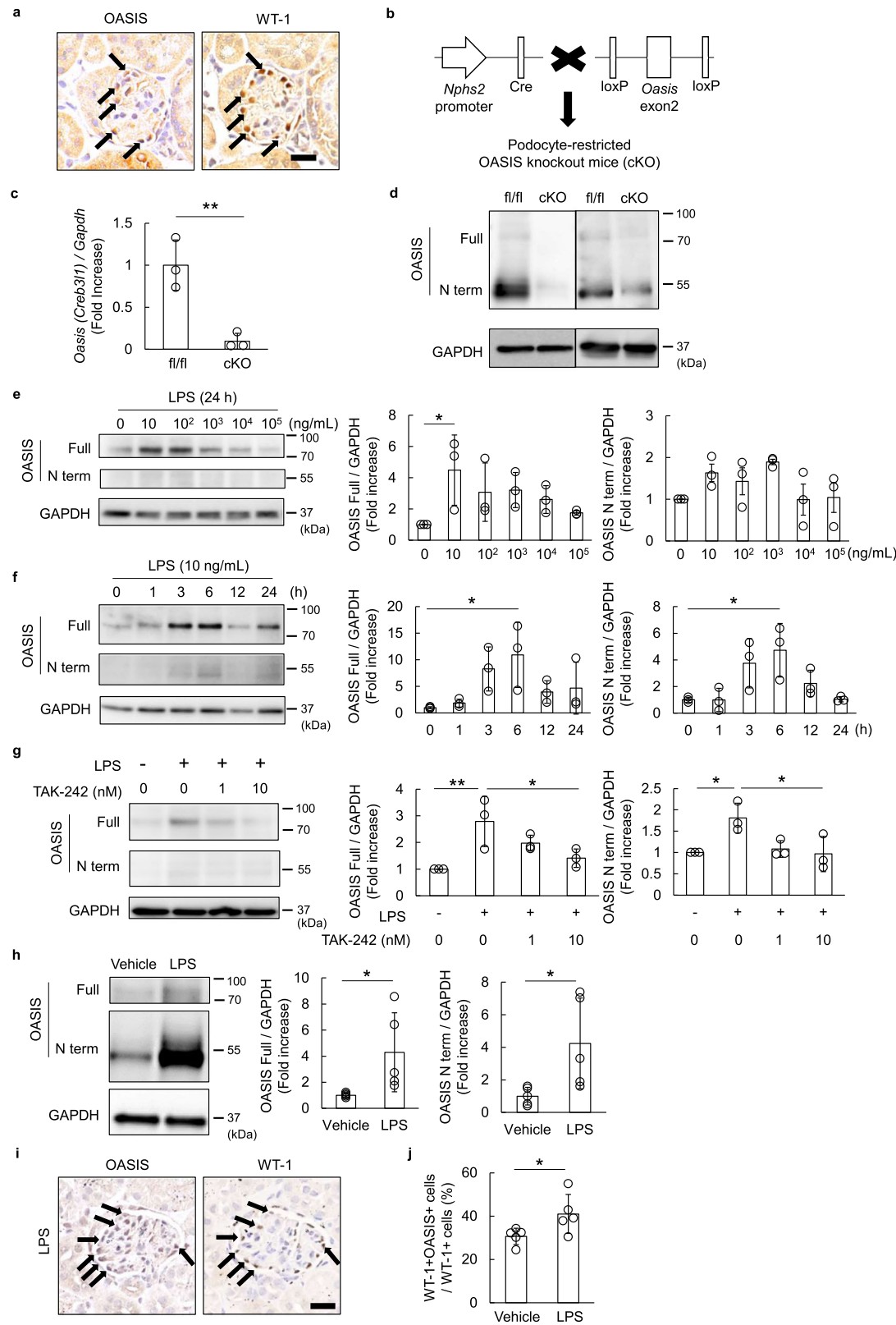

protein expression of OASIS increased in the glomeruli of the murine DN model (Fig. 4a). Next, the OASIS cKO and control mice were subjected to uninephrectomy, to accelerate the progression of kidney injury, followed by injection of STZ. We initially confirmed that STZ treatment blunted body weight gain, while increasing blood glucose level and urine volume, in both the OASIS cKO and control mice (Fig. 4b and Supplementary Fig. 3). Intriguingly, the

DN-increased kidney weight/body weight was suppressed in the OASIS cKO mice, as compared to that in the control mice (Fig. 4c). Consistent with the LPS model, although podocyte OASIS deficiency did not affect the 24-hour-urinary albumin excretion and ACR, the sCr level was significantly reduced in the OASIS cKO mice (sCr: control, $0.74 \pm 0.10$ mg/dL; cKO, $0.57 \pm 0.09$ mg/dL, $P < 0.01$) (Fig. 4d–f). Additionally, the transcript expression of podocyte

**Fig. 1 LPS treatment increased OASIS expression in podocytes. a** Serial sections of murine kidneys were analyzed with the help of immunohistochemical staining, using an anti-OASIS or anti-WT-1 antibody. Nuclei were stained with hematoxylin. Representative images are shown. Arrows: OASIS-expressing WT-1-positive cells. Scale bar: 20 μm. **b** Schematic for the generation of podocyte-restricted OASIS knockout (cKO) mice. *Oasis* fl/fl mice were used as control. **c**, **d** Glomeruli were isolated from the kidneys of fl/fl and cKO mice. The transcript expression of *Oasis/Creb3l1* was examined using quantitative PCR (**c**). Data are shown as mean ± SD ($n = 3$ for each group), **$P < 0.01$, as analyzed using Student's *t*-test. Immunoblotting was performed using anti-OASIS and anti-GAPDH antibodies (**d**). **e**, **f** Murine-cultured podocytes were stimulated with LPS at the indicated concentrations, for 24 h (**e**), or with LPS (10 ng/mL) at the indicated time-points (**f**). Immunoblotting was performed using anti-OASIS and anti-GAPDH antibodies. Representative images and quantitative analysis for OASIS expression levels are shown. Data are shown as mean ± SD ($n = 3$ for each group), *$P < 0.05$, as analyzed using Dunnett test. **g** Murine-cultured podocytes were pre-treated with TAK-242, a TLR-4 inhibitor, at the indicated concentrations, for 30 min, before stimulation with LPS (10 ng/mL). Six hours after LPS stimulation, immunoblotting was performed using anti-OASIS and anti-GAPDH antibodies. Representative images and quantitative analysis for OASIS expression levels are shown. Data are shown as mean ± SD ($n = 3$ for each group), *$P < 0.05$ and **$P < 0.01$, as analyzed using Dunnett test. **h** Glomeruli were isolated from kidney sections of vehicle- or LPS-treated mice using LCM. Immunoblotting was performed using anti-OASIS and anti-GAPDH antibodies. Representative images and quantitative analysis for OASIS expression levels are shown. Data are shown as mean ± SD ($n = 5$ for each group), *$P < 0.05$, as analyzed using Student's *t*-test. **i**, **j** Immunohistochemical staining was performed using an anti-OASIS or anti-WT-1 antibody, after LPS treatment. Representative images (**i**) and quantitative measurement of WT-1+OASIS+ cells (**j**) are shown. Arrows: WT-1- and OASIS-positive cells. Scale bar: 20 μm. Data are shown as mean ± SD ($n = 5$ for each group), *$P < 0.05$, as analyzed using Student's *t*-test.

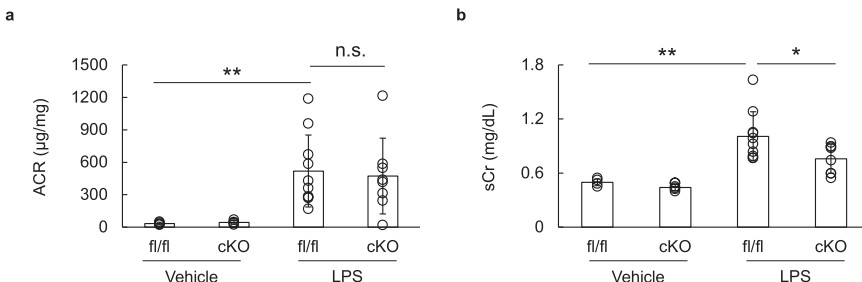

**Fig. 2 Podocyte-restricted OASIS deficiency attenuated LPS-induced acute kidney dysfunction. a**, **b** OASIS cKO and fl/fl mice were intraperitoneally administered with LPS, at a concentration of 10 mg/kg. Urinary albumin creatinine ratio (ACR) and serum creatinine (sCr) levels were measured 24 h after LPS injection. Data are shown as mean ± SD ($n = 5$ for fl/fl-vehicle and cKO-vehicle, $n = 10$ for fl/fl-LPS, and $n = 8$ for cKO-LPS), *$P < 0.05$ and **$P < 0.01$, as analyzed using Dunnett test.

markers and podocyte foot process width were unaltered upon podocyte OASIS knockout (Fig. 4g–j). Meanwhile, tubular injury, including tubular dilatation, was significantly suppressed in the OASIS cKO mice (Fig. 4k, l). LTL-positive area was preserved in the OASIS cKO mice, as compared to that in the control mice (Fig. 4m, n). These data suggested that deletion of podocyte OASIS also suppressed kidney dysfunction in the STZ-induced DN model, concomitant with a decrease in tubular damage.

**Protein kinase C iota (PKCι, Prkci) mediated tubular protection in the OASIS cKO mice, after LPS treatment.** Next, we investigated the molecular mechanisms underlying podocyte OASIS deletion-mediated tubular protection. The murine podocyte cell line was transfected with a lentivirus expressing the active form of *Oasis* or *venus* (as a control), followed by microarray analysis (Fig. 5a, b). Here, we focused on genes whose proteins could be secreted into the extracellular spaces, thereby mediating the interaction of podocytes with tubular epithelial cells, and found that there was a decrease in the expression of 5 annotated genes, while the expression of 17 annotated genes increased in the OASIS-overexpressing podocytes. Moreover, the expression of these 22 genes in the podocytes was measured under conditions of elevated OASIS upon LPS treatment (Fig. 5c). Upon doing so, the transcript expression of *Prkci*, *Wnt5a*, and *Gadd45α*, which was negatively regulated by OASIS overexpression, decreased significantly at 6 h after LPS treatment, as analyzed using quantitative PCR, while that of the others was not changed or detected. Furthermore, when the transcript expression of *Prkci*, *Wnt5a*, and *Gadd45α* in the kidneys of LPS-treated OASIS cKO mice was examined, among the three genes, only that of *Prkci* increased, compared to those in the kidneys of control

mice (Fig. 5d–f). Immunostaining revealed that PRKCI expression was observed in the glomerular cells, including podocytes, as well as in the tubulointerstitial space (Supplementary Fig. 4). The intensity of PRKCI staining in the glomeruli tended to increase in the OASIS cKO mice, though the change was not statistically significant. Since PRKCI can be secreted extracellularly, it is likely that no clear difference in PRKCI expression was observed in the kidney tissues of control and OASIS cKO mice. In fact, immunoblotting demonstrated that the protein expression of PRKCI increased in the urine of OASIS cKO mice (Fig. 5g), suggesting that PRKCI serves as a candidate effector that mediates the interaction of podocytes with the tubular epithelial cells in the kidneys of OASIS cKO mice. PRKCI is a member of atypical protein kinase C[29] that is known to serve as a key player in the establishment of cell polarity[30,31]. To determine the effects of PRKCI on tubular injury, isolated murine tubular cells were treated with recombinant PRKCI, followed by stimulation with LPS (Fig. 5h). Quantitative PCR revealed that LPS-induced upregulation of *Lcn2* mRNA expression was suppressed upon PRKCI treatment. To confirm that OASIS exhibits detrimental effects on tubular cells, mechanistically through PRKCI, the OASIS cKO mice were treated with a PRKCI inhibitor after LPS injection. Significantly, the tubular-protective effect in OASIS cKO mice was abolished upon PRKCI inhibitor treatment (Fig. 5i–k). Taken together, the PRKCI secreted from OASIS knockout podocytes played a protective role against LPS-induced tubular injury.

**Podocyte-restricted overexpression of the active form of OASIS leads to severe kidney injury.** To reinforce the contribution of podocyte OASIS to the disruption of kidney homeostasis, we

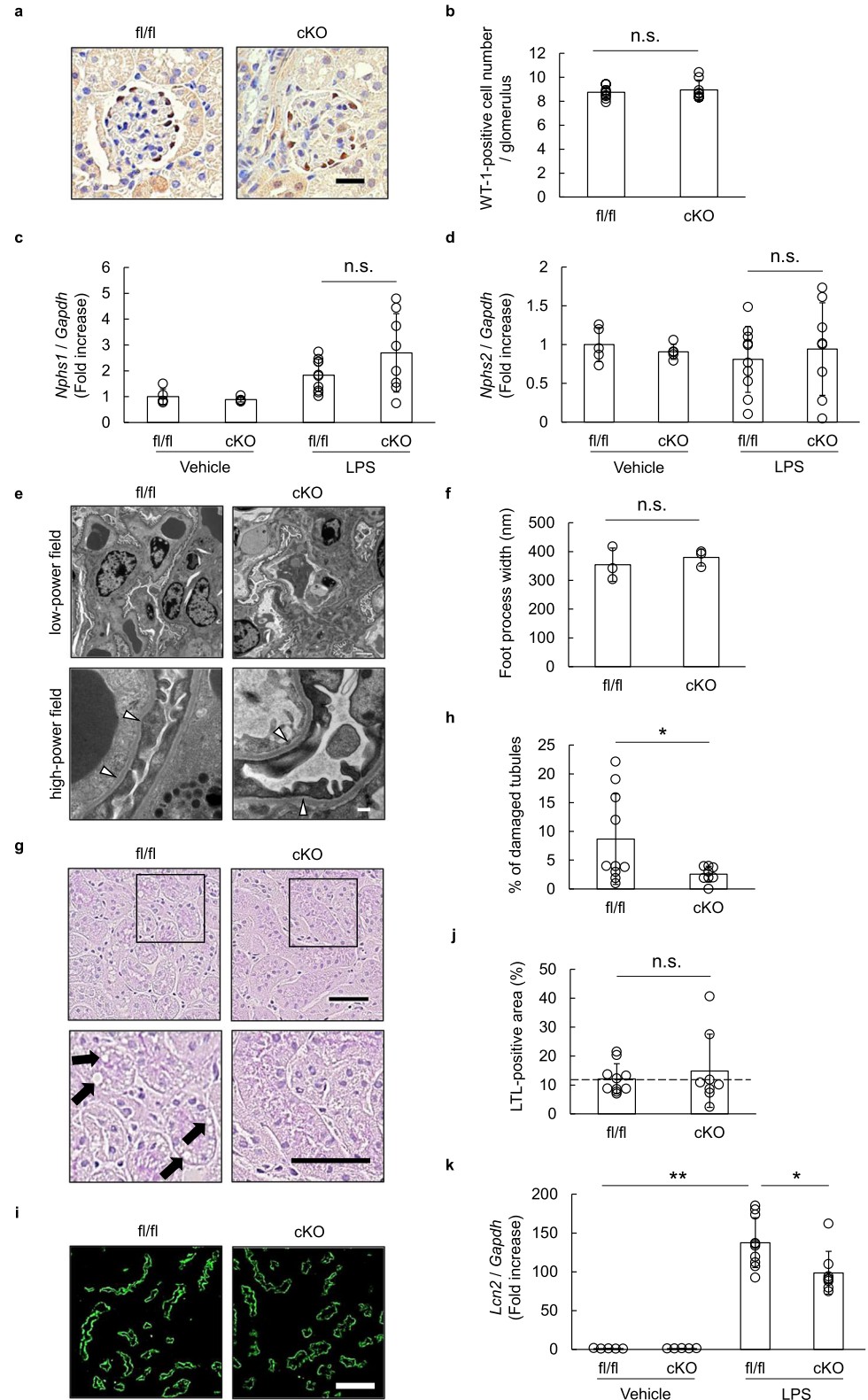

generated mice overexpressing the podocyte-restricted active form of OASIS (OASIS TG) (Fig. 6a, b). First, OASIS overexpression in the glomeruli was confirmed using LCM and immunoblotting (Fig. 6c). Podocyte-restricted overexpression of the active form of OASIS resulted in albuminuria (ACR at 12 weeks of age: control, 35.1 ± 24.9 µg/mg; TG, 8930.0 ± 8461.4 µg/mg, $P = 0.007$), while the sCr level was not significantly changed (Fig. 6d, e). Moreover, electron microscopic

analysis indicated that OASIS TG showed significant podocyte foot process effacement (Fig. 6f). Consistently, the in vitro study demonstrated that sustained overexpression of OASIS in the podocytes resulted in abnormalities in the cytoskeleton of podocytes (Supplementary Fig. 5). In addition, tubular damage was observed and there was an increase in the tubular injury marker *Lcn2* mRNA in the kidneys of OASIS TG mice (Fig. 6g–i). Masson's trichrome staining and quantitative PCR for fibrosis

**Fig. 3 Ablation of podocyte OASIS suppressed LPS-induced tubular injury. a, b** Twenty-four hours after LPS treatment, immunohistochemical analysis was performed using an anti-WT-1 antibody. Representative images are shown (**a**). Scale bar: 20 μm. The number of WT-1-positive cells was counted (**b**). Data are shown as mean ± SD ($n = 10$ for fl/fl-LPS and $n = 8$ for cKO-LPS). **c, d** The transcript expression of *Nphs1* and *Nphs2* was examined using quantitative PCR. The expression of the transcripts was normalized to that of *Gapdh*. Data are shown as mean ± SD ($n = 5$ for fl/fl-vehicle and cKO-vehicle, $n = 10$ for fl/fl-LPS, and $n = 8$ for cKO-LPS). **e, f** Representative transmission electron microscopy images are shown at low and high magnifications. The arrowheads indicate podocyte foot process effacement. Scale bar: 2 μm (low-power field) and 200 nm (high-power field). Podocyte foot process width was measured (**f**). Data are shown as mean ± SD ($n = 3$ for each group). **g–j** Kidney sections were stained with Periodic acid–Schiff (**g**, **h**) or with LTL (**i**, **j**). Representative images are shown. Arrows indicate LPS-induced vacuolization in tubular cells. Scale bar: 50 μm. Quantitative assessment of tubular injury (**h**) and LTL-positive area (**j**). Data are shown as mean ± SD ($n = 10$ for fl/fl-LPS and $n = 8$ for cKO-LPS), *$P < 0.05$, as analyzed using Student's $t$-test. The dashed line indicates the level of LTL-positive area in the non-treated mice. **k** The transcript expression of *Lcn2* was examined using quantitative PCR. The expression of the transcripts was normalized to that of *Gapdh*. Data are shown as mean ± SD ($n = 5$ for fl/fl-vehicle and cKO-vehicle, $n = 10$ for fl/fl-LPS, and $n = 8$ for cKO-LPS), *$P < 0.05$ and **$P < 0.01$, as analyzed using Dunnett test.

markers, including collagen1a1 and fibronectin, showed that significant kidney fibrosis occurred in the OASIS TG mice, as compared to that in the control mice (Fig. 6j–m). Interestingly, the OASIS TG mice showed lower survival rate and body weight than the control mice (Supplementary Fig. 6). By the age of 8 weeks, mice with ACR levels exceeding 15,000 μg/mg were defined as the severe group, and their kidney function was evaluated at that time-point (Supplementary Fig. 7). sCr level was higher in the severe group, as compared to those in the control and OASIS TG groups that survived to 12 weeks of age. The increased expression of *Lcn2* mRNA was more pronounced in the severe group, suggesting that in addition to glomerular damage, the severity of tubular injury in the TG group may be related to the decreased kidney function and survival rate. We confirmed another line of podocyte-restricted OASIS TG mouse that showed similar phenotypic changes (Supplementary Fig. 8). Collectively, upregulation of podocyte OASIS leads to severe kidney injury.

**OASIS-positive podocytes were increased in the glomeruli of patients with minimal change nephrotic syndrome (MCNS) and DN.** Finally, we assessed the relevance of OASIS expression in human kidney diseases. As described above, the expression of OASIS was upregulated in the glomeruli of LPS-treated and STZ-induced DN mice. The LPS-induced kidney injury model is used not only as an acute kidney injury model, but also as an MCNS model[32,33]. Thus, we prepared human biopsy samples from MCNS and DN patients, and performed immunohistochemistry in the serial sections of the kidneys using an anti-OASIS or anti-WT-1 antibody. Consistent with the murine kidneys, OASIS-positive podocytes were also detected in the human kidney samples (Fig. 7). Importantly, the ratio of podocytes with detectable OASIS increased in the glomeruli of the MCNS and DN patients, as compared to those of the control, indicating that upregulation of OASIS could be associated with the pathogenesis of MCNS and DN.

## Discussion

In the present study, we demonstrated that OASIS deletion in podocytes attenuated LPS- and STZ-induced kidney dysfunction, concomitant with decreased tubular injury, at least in part by the induction of PRKCI. Moreover, OASIS overexpression in the podocytes showed tubular injury and tubulointerstitial fibrosis, with severe albuminuria and podocyte degeneration. Finally, OASIS-positive podocytes increased in the kidneys of MCNS and DN patients. These data identified that the upregulation of OASIS in podocytes impaired kidney homeostasis. Our findings provide the first demonstration of the role of OASIS in podocyte homeostasis.

In this study, LPS induced OASIS in a TLR4-dependent manner. OASIS was more remarkably activated by LPS at lower doses, as compared to higher doses. Therefore, the induction of OASIS might depend on the kinetics of LPS signaling. Indeed, it has been reported that the signal transduction mediated by low-dose LPS differs from that mediated by high-dose LPS[34,35]. For example, Maitra et al. suggested that low-dose LPS activates interleukin 1 receptor associated kinase (IRAK)-1/ATF2 signaling and suppresses phosphatidylinositol 3-kinase (PI3K), IRAK-M, mitogen-activated protein kinase phosphatase 1, and v-rel avian reticuloendotheliosis viral oncogene homolog B, resulting in induction of mild and prolonged expression of pro-inflammatory mediators[34]. Meanwhile, high-dose LPS activated not only IRAK4/2/1/nuclear factor-κB, but also the PI3K pathway, and caused robust transient expression of pro/anti-inflammatory mediators. Thus, it could be possible that the difference in signaling pathways between the low and high concentrations of LPS may affect the induction of OASIS.

Podocyte dysfunction and tubular/tubulointerstitial injury were observed in the OASIS TG mice, while only tubular injury, but not podocyte dysfunction, was suppressed in the OASIS cKO mice, after LPS or STZ treatment. The differential effect of OASIS on podocyte dysfunction between the two genetically engineered animals might be explained in terms of the levels and/or persistency of OASIS expression. It is noteworthy that the ratio of OASIS-positive podocytes increased in the glomeruli of MCNS and DN patients, suggesting that persistent expression of OASIS results in podocyte dysfunction. Podocyte dysfunction is closely associated with cytoskeletal abnormality. Interestingly, abnormalities in the cytoskeleton were more severe at day 4 after OASIS overexpression, as compared to that at day 2, in cultured podocytes (Supplementary Fig. 5), suggesting that OASIS induces cytoskeletal alteration depending on its expression and/or persistent levels. Another explanation is that the biological function of OASIS might be modified by the other signals in the LPS or STZ model. It has been reported that OASIS interacts with other factors, such as Smad4[10] and hypoxia-inducible factor-1α[12], to function. Since various signaling molecules play crucial roles in response to LPS, as described above, podocyte OASIS may regulate the pathogenesis of kidney diseases through interactions with some component, under pathological conditions.

Much focus has been placed on the interplay between podocytes and tubular epithelial cells in the pathogenesis of kidney diseases. For example, tubular epithelial cells communicate with podocytes through sirtuin 1 and nicotinic acid metabolism, in the early stage of DN[5]. Conversely, podocyte-derived extracellular vesicles maintain the functions of renal tubular cells[6]. In this study, since tubular injury, but not podocyte injury, was attenuated in the OASIS cKO mice after LPS treatment, we paid attention to the podocytes-tubular epithelial cells interaction via podocyte-derived downstream effectors of OASIS, and identified PRKCI (PKCι) as a protective factor against tubular injury, which was negatively regulated by OASIS in podocytes. Huber et al.

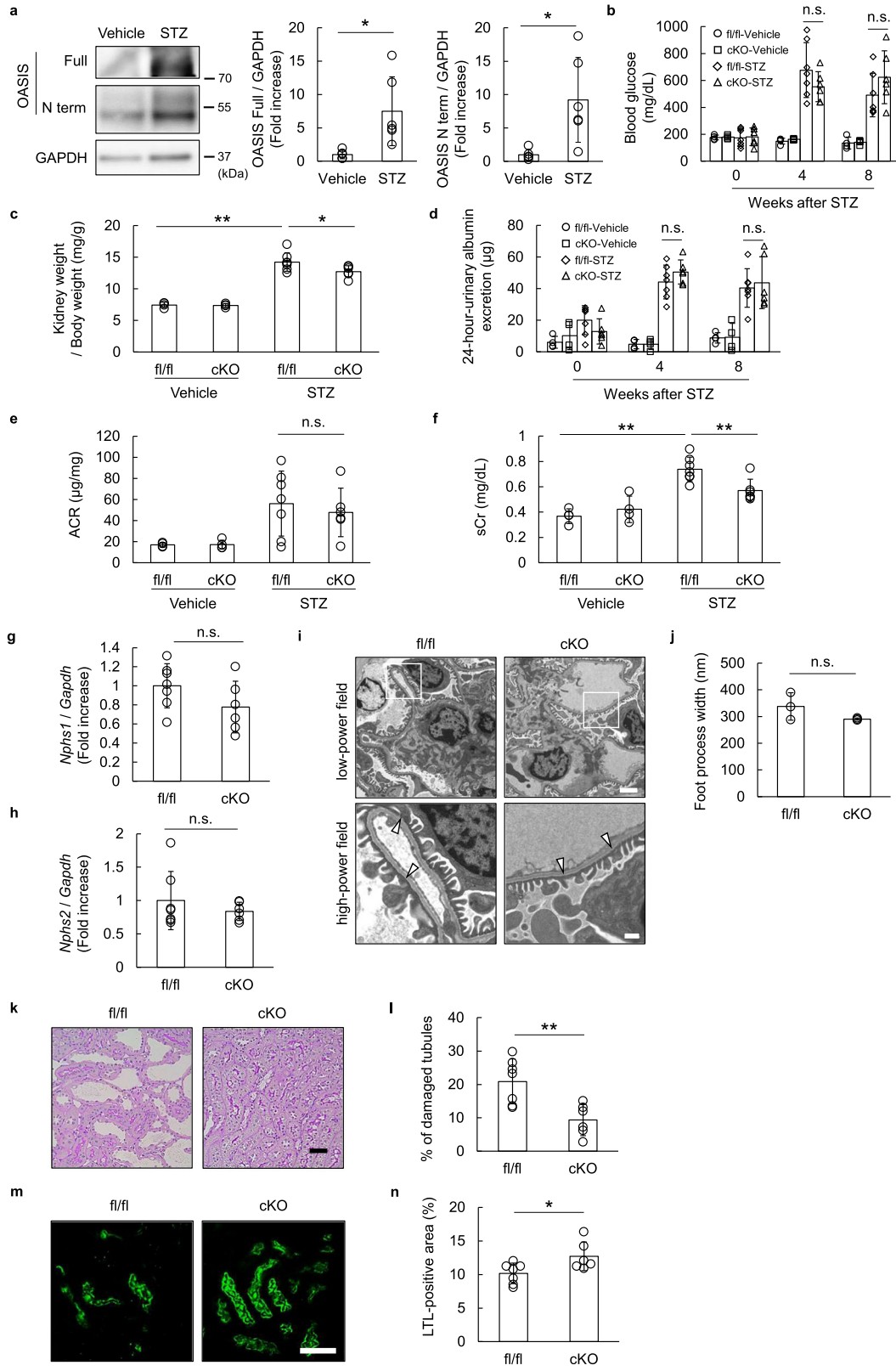

demonstrated that knockout of PKCλ/ι in podocytes causes nephrotic syndrome[36,37]. Moreover, deletion of PKCλ/ι exhibited a severe actin cytoskeletal phenotype in cultured podocytes[38]. Additionally, PRKCI reportedly suppresses tissue-damage in colitis[39]. Combined with our data, it is conceivable that PRKCI plays a protective role in kidney diseases. Consistent with this result, the protein level of PRKCI was higher in the urine of

OASIS cKO mice than in that of control mice, after LPS treatment. Furthermore, PRKCI inhibitor abolished the tubular-protective effect in OASIS cKO, thereby showing that podocyte-secreted PRKCI acted on the luminal side of tubular epithelial cells and protected against tubular injury. Incidentally, since creatinine is known to be secreted from tubules at the rate of 50% in male mice[40], we considered that LPS-induced tubular injury

**Fig. 4 Podocyte-restricted OASIS deletion ameliorated kidney dysfunction in the STZ-induced diabetic nephropathy model. a** C57BL/6 mice were intraperitoneally administered with STZ. Glomeruli were isolated from the kidney sections of vehicle- or STZ-treated mice, at 8–12 weeks after STZ, using LCM. Immunoblotting was performed using anti-OASIS and anti-GAPDH antibodies. Representative images and quantitative analysis for OASIS expression levels are shown. Data are shown as mean ± SD ($n = 5$ for vehicle and $n = 6$ for STZ), *$P < 0.05$, as analyzed using Student's $t$-test. **b** OASIS cKO and fl/fl mice were subjected to uninephrectomy, followed by treatment with STZ or vehicle, as a control. Blood glucose levels were measured at 0, 4, and 8 weeks after STZ treatment. Data are shown as mean ± SD ($n = 4$ for fl/fl-vehicle and cKO-vehicle, $n = 7$ for fl/fl-STZ, and $n = 6$ for cKO-STZ). **c** Ratio of kidney weight to body weight was calculated at 8 weeks after STZ treatment. Data are shown as mean ± SD ($n = 4$ for fl/fl-vehicle and cKO-vehicle, $n = 7$ for fl/fl-STZ, and $n = 6$ for cKO-STZ), *$P < 0.05$ and **$P < 0.01$, as analyzed using Dunnett test. **d** Twenty-four-hour-urinary albumin excretion was measured at 0, 4, and 8 weeks after STZ treatment. **e** ACR was measured at 8 weeks after STZ treatment. **f** sCr level was measured at 8 weeks after STZ treatment. Data are shown as mean ± SD ($n = 4$ for fl/fl-vehicle and cKO-vehicle, $n = 7$ for fl/fl-STZ, and $n = 6$ for cKO-STZ), **$P < 0.01$, as analyzed using Dunnett test. **g, h** The transcript expression levels of *Nphs1* and *Nphs2* were examined at 8 weeks after STZ treatment, using quantitative PCR. The expression of the transcripts was normalized to that of *Gapdh*. Data are shown as mean ± SD ($n = 7$ for fl/fl-STZ and $n = 6$ for cKO-STZ). **i** Representative transmission electron microscopy images are shown at low and high magnifications. The arrowheads indicate podocyte foot process effacement. Scale bar: 2 µm (low-power field) and 500 nm (high-power field). **j** Podocyte foot process width was examined. Data are shown as mean ± SD ($n = 3$ for each group). **k–n** Eight weeks after STZ treatment, the kidney sections were stained with Periodic acid–Schiff (**k, l**) or LTL (**m, n**). Representative images are shown. Scale bar: 50 µm. Quantitative assessment of tubular injury (**l**) and LTL-positive area (**n**). Data are shown as mean ± SD ($n = 7$ for fl/fl-STZ and $n = 6$ for cKO-STZ), *$P < 0.05$, as analyzed using Student's $t$-test.

was attenuated in the OASIS cKO mice, resulting in suppression of the sCr level.

Furthermore, as OASIS was also detected in tubular cells according to our previous report[17], the Human Protein Atlas database (https://www.proteinatlas.org/), and the immunohistochemical data in Fig. 7, further studies are needed to examine the roles of OASIS in tubular cells using cell-specific OASIS knockout mice.

This study does have some limitations. Concretely, OASIS protein in the in vivo kidney sections was not specifically detected using immunofluorescence, as described previously[17]. Thus, we could not clearly indicate the co-localization of OASIS protein and podocyte markers. In the present study, we performed immunohistochemistry using serial sections and used podocyte-specific knockout mice to prove the presence of OASIS in podocytes. Further studies are needed to develop other detection systems for OASIS expression in kidneys, like green fluorescence protein-fused OASIS-expressing mice. Although 15–30% of podocytes expressed nuclear OASIS protein under physiological conditions, in both human and animal samples, the physiological function of OASIS remains unclear, since podocyte-restricted OASIS-deficient mice did not show any obvious abnormality during the experimental period. Additionally, although the number of podocytes with detectable OASIS increased under the human and mouse renal pathological conditions, it is not known whether OASIS was increased or activated in each podocyte in the in vivo model. Another limitation is that the understanding of renal function in the murine model is not necessarily consistent with that in the clinical setting. For example, creatinine is secreted from tubules at the rate of 50% in male mice[40]. Therefore, sCr concentration might be influenced by both glomerular and tubular function in mice, while it is influenced preferentially by only glomerular function in humans.

In conclusion, podocyte OASIS evoked tubular damage through downregulation of PRKCI after LPS treatment and/or podocyte injury in mice. OASIS upregulation in podocytes contributes to the disruption of kidney homeostasis. Further investigation on the pathophysiological roles of podocyte OASIS in various kidney diseases could provide insight into the possibility of application of OASIS as a therapeutic target for kidney diseases.

## Methods
**Animal study.** Care of animals was performed according to the Osaka University Animal Care Guidelines and RIKEN Regulations for Animal Experiments. All experimental procedures conformed to the Guide for the Care and Use of

Laboratory Animals, Eighth Edition, updated by the US National Research Council Committee in 2011, and were approved by the Animal Care and Use Committee at the Graduate School of Pharmaceutical Sciences, Osaka University (approval number: Douyaku 28-14, R01-1) and the Institutional Animal Care and Use Committee of RIKEN Kobe Branch. The mice were maintained in a 12 h/12 h light/dark cycle, and received food and water ad libitum at the Animal Care Facility of the Graduate School of Pharmaceutical Sciences, Osaka University and RIKEN BDR Kobe Branch. At the end-points of all the experiments, the mice were deeply anesthetized with isoflurane, following which their blood and kidneys were collected. All efforts were made to minimize suffering.

**Generation of podocyte-restricted OASIS knockout or overexpressing transgenic mice.** In order to generate podocyte-restricted OASIS knockout (cKO) mice (male, 8–10-week-old) using the Cre-loxP system, *Oasis* fl/fl mice[17] were mated with *Neph2*-Cre mice[41,42]. *Oasis* fl/fl mice were used as the control.

To establish mice overexpressing the podocyte-restricted active form of OASIS (TG; male, 4–12-week-old), we generated CAG/CATZ/OASIS mice (accession no. CDB0546T: http://www2.clst.riken.jp/arg/TG%20mutant%20mice%20list.html). These CAG/CATZ/OASIS mice were mated with *Neph2*-Cre mice. *Neph2*-Cre mice were used as control. Two lines of OASIS TG were analyzed in this study.

**Generation of the LPS-induced kidney injury model and treatment with PRKCI inhibitor.** C57BL/6 J male mice (8–10-week-old) were purchased from Japan SLC (Shizuoka, Japan). The mice were treated with a single intraperitoneal injection of 10 mg/kg body weight LPS (O111:B4, Sigma–Aldrich/Merck, Darmstadt, Germany). Twenty-four hours after the injection, the kidneys were perfused with phosphate-buffered saline (PBS) and then collected.

PRKCI inhibitor 1 was purchased from MedChemExpress (HY-126146, Monmouth Junction, NJ, USA) and dissolved in corn oil (Sigma–Aldrich/Merck). fl/fl and cKO mice were treated with the PRKCI inhibitor 1 (10 mg/kg), at 1 and 12 h after LPS injection. Twenty-four hours after the LPS injection, the kidneys of the mice were collected.

**DN model.** C57BL/6 J male mice (8–10-week-old) were treated with an intraperitoneal injection of 50 mg/kg body weight STZ (Sigma-Aldrich/Merck) diluted in 0.1 mol/L citrate buffer (pH 4.5), for 5 consecutive days. To accelerate the progression of kidney injury, nephrectomy of the left kidney was performed in the OASIS cKO and control mice, one week before STZ treatment.

**Cell culture.** The immortalized murine podocyte cell line was a gift from Dr. Peter Mundel and differentiated under nonpermissive conditions (10–14 days at 37 °C, no interferon-γ)[43]. For LPS treatment, differentiated podocytes were cultured in RPMI-1640 medium containing 1% fetal bovine serum (FBS). TAK-242 (Cayman Chemical Company, Ann Arbor, MI, USA) was used to inhibit the TLR4 signaling.

Glomeruli were isolated from the mice with reference to previous report[44]. Specifically, $4 \times 10^8$ Dynabeads™ M-450 Tosylactivated (Invitrogen/Thermo Fisher Scientific, Waltham, MA, USA) diluted in 40 mL of PBS were perfused through the heart. The kidneys were harvested, minced into small pieces, and digested in collagenase solution [1 mg/mL collagenase A (Roche/Merck, Darmstadt, Germany) and 100 U/mL DNase1 (Roche)], at 37 °C for 60 min. The digested tissues were filtered through a 100 µm cell strainer, following which the cell suspensions were centrifuged at $200 \times g$ for 5 min. The supernatant was discarded and the cell pellet was re-suspended in 1 mL Hanks' Balanced Salt Solution (HBSS). After that, the glomeruli containing Dynabeads™ were gathered using a Cell Separation Magnet

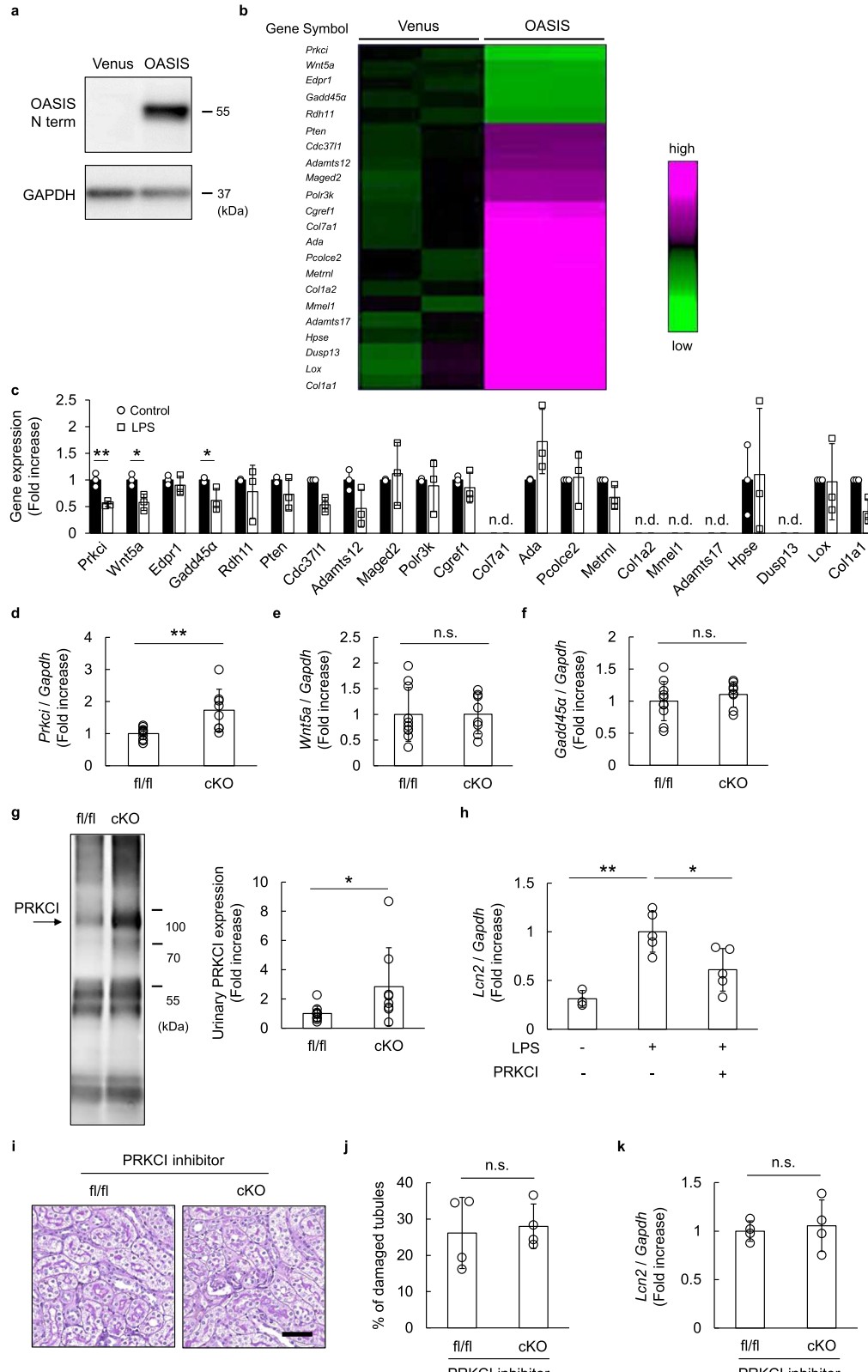

(BD Biosciences, San Jose, CA, USA) and washed three times with HBSS. FastGene RNA Premium Kit (Nippon Genetics, Tokyo, Japan) was used for total RNA extraction from the isolated glomeruli, according to the manufacturer's protocol.

Isolation of kidney tubules was performed with reference to previous reports[45,46]. Specifically, the renal cortex harvested from C57BL/6 J male mice (8–10-week-old) was sliced into small pieces. The fragments were then digested with dissection solution containing 1 mg/mL collagenase A and 100 µg/mL trypsin inhibitor (Sigma), at 37 °C for 30 min. After digestion, the tissues were sieved through

two nylon sieves (pore size 250 and 70 µm). Tubular fragments retained in the 70 µm sieve were re-suspended in HBSS, following which the suspensions were centrifuged at $170 \times g$ for 5 min. The obtained supernatant was discarded and the cell pellet was re-suspended in culture media (DMEM/Ham's F-12, supplemented with 15 mmol/L HEPES, 2 mmol/L L-glutamine, 50 nmol/L hydrocortisone, 5 µg/mL insulin, 5 µg/mL transferrin, 50 nmol/L selenium, 0.55 mmol/L sodium pyruvate, non-essential amino acids, penicillin, and streptomycin), and seeded onto collagen type I-coated cell culture plates. The isolated tubular cells were treated with recombinant PRKCI

**Fig. 5 PRKCI, a candidate downstream of OASIS in podocytes, suppressed LPS-induced tubular injury. a, b** Murine cultured podocytes were transfected with a lentivirus expressing the active form of *Oasis* or *venus* (as a control). Forty-eight hours after transfection, immunoblotting was performed using anti-OASIS and anti-GAPDH antibodies. Representative images are shown (**a**). **b** DNA microarray analysis was performed. Heat-map shows the genes, whose protein can be secreted into the extracellular spaces, with >2-fold change in OASIS-overexpressed cells, and a *q*-value of <0.1. **c** Murine podocytes were treated with LPS for 6 h. The transcript expression of candidate downstream genes of OASIS was examined using quantitative PCR. The expression of the transcripts was normalized to that of *Gapdh*. Data are shown as mean ± SD ($n = 3$ for each group), *$P < 0.05$ and **$P < 0.01$, as analyzed using Student's *t*-test. **d**–**f** The transcript expression of *Prkci*, *Wnt5a*, and *Gadd45a* was examined in the kidneys of LPS-treated fl/fl and cKO mice, using quantitative PCR. The expression of the transcripts was normalized to that of *Gapdh*. Data are shown as mean ± SD ($n = 10$ for fl/fl-LPS and $n = 8$ for cKO-LPS), **$P < 0.01$, as analyzed using Student's *t*-test. **g** Spot urine samples were collected from the LPS-treated fl/fl and cKO mice. Equal volume of urine (4 μL) was used for immunoblotting. Representative images and quantitative analysis for PRKCI expression levels are shown. Data are shown as mean ± SD ($n = 10$ for fl/fl-LPS and $n = 8$ for cKO-LPS), *$P < 0.05$, as analyzed using Student's *t*-test. **h** Isolated murine tubular cells were pre-treated with recombinant PRKCI, at a concentration of 300 ng/mL, for 1 h, followed by stimulation with LPS (10 μg/mL). Twenty-four hours after LPS stimulation, the transcript expression of *Lcn2* was examined using quantitative PCR. Data are shown as mean ± SD [$n = 3$ for LPS(-)PRKCI(-), $n = 5$ for LPS(+)PRKCI(-) and LPS(+)PRKCI(+)], *$P < 0.05$ and **$P < 0.01$, as analyzed using Dunnett test. **i**–**k** fl/fl and cKO mice were treated with PRKCI inhibitor 1, at a concentration of 10 mg/kg, at 1 and 12 h after LPS injection. Twenty-four hours after LPS injection, the kidney sections were stained with Periodic acid–Schiff. Representative images (**i**) and quantitative assessment of tubular injury (**j**) are shown. Scale bar: 50 μm. The mRNA expression of *Lcn2* was quantitatively measured (**k**). Data are shown as mean ± SD ($n = 4$).

(Invitrogen, 300 ng/mL) for 1 h, followed by stimulation with LPS (10 μg/mL) for 24 h.

**Preparation of cytosolic and nuclear fractions from podocytes**. Podocytes were stimulated with LPS (10 ng/mL) for 6 h. After LPS treatment, the cells were scraped with PBS and centrifuged at 3000 rpm for 5 min. The pellets were then gently suspended in hypotonic buffer and incubated on ice for 15 min. After incubation, NP-40 was added to the cells and they were centrifuged at 3000 rpm for 5 min. The supernatant was used as the cytosolic fraction, while the pellet was re-suspended in nuclear extraction buffer. After incubation on ice for 30 min, the homogenates were centrifuged at $14,000 \times g$ for 10 min. The obtained supernatant was then used as the nuclear fraction. Equal amounts of proteins from cytosolic and nuclear fractions were used for immunoblotting.

**LCM**. Murine kidneys were frozen in O.C.T. Compound (Sakura Finetek Japan, Tokyo, Japan). The sections (10 μm-thick) were prepared using Leica CM 1950 (Leica, Wetzlar, Germany) and attached to PEN Membrane (Leica). After fixing with ethanol, the nuclei were stained with hematoxylin. Approximately 350 glomeruli for each sample were isolated and collected using LMD7000 (Leica). The samples were then subjected to immunoblotting.

**Immunoblotting**. Immunoblot analysis was performed with reference to previous report[17]. Proteins were separated by SDS-PAGE and transferred onto polyvinylidene difluoride membrane (Millipore/Merck, Darmstadt, Germany). The membrane was blocked with Tris-buffered saline (TBS)-0.05% Tween20 (Nacalai Tesque, Kyoto, Japan) containing 2% skim milk (Becton, Dickinson and Company, New Jersey, USA) or 5% bovine serum albumin (Nacalai Tesque) for 1 h, followed by incubation with the primary antibody diluted with blocking buffer or Can Get Signal Immunoreaction Enhancer Solution (TOYOBO, Osaka, Japan) overnight at 4 °C. Anti-OASIS (1:500; MABE1017; Millipore/Merck), anti-Lamin B1 (1:10000; 3C10G12; ProteinTech, Chicago, IL, USA), anti-PRKCI (1:1000; 610175; BD Biosciences), and anti-GAPDH (1:1000; MAB374; Millipore) antibodies were used as primary antibodies. After washing with TBS-T, the membrane was incubated with the secondary antibody for 1 h at room temperature. Horseradish peroxidase (HRP)-conjugated goat anti-mouse IgG (1:2000; Jackson ImmunoResearch, West Grove, PA, USA) and HRP-conjugated goat anti-rabbit IgG (1:2000; Cell Signaling Technology, Danvers, MA, USA) were used as secondary antibodies. The HRP activity was detected with ECL Western Blotting Substrate (Promega, Madison, WI, USA) or Chemi-Lumi One Super (Nacalai Tesque), and visualized with ImageQuant LAS 4010 (GE Healthcare, Chicago, IL, USA). Band intensities were quantified using ImageJ software (National Institutes of Health, Bethesda, MD, USA).

**Immunohistochemistry/immunofluorescence**. The kidney tissues were fixed in 4% paraformaldehyde for 24 h, and then embedded in paraffin, followed by sectioning to a thickness of 3 μm. The serial sections were stained with anti-OASIS (1:200; AF4080; R&D systems, Minneapolis, MN, USA), anti-WT-1 (1:500; sc-393498; Santa Cruz Biotechnology, Dallas, TX, USA), or anti-PRKCI (1:200; 610175; BD Biosciences) antibody, for the detection of podocytes using the VECTASTAIN® ABC kit (Vector Laboratories, Burlingame, CA, USA). The nuclei were also stained with hematoxylin. For LTL staining, the kidney sections were stained with FITC-conjugated LTL (1:150; Vector Laboratories), while the nuclei were stained with DAPI (1:200). Images were taken using a BZ-X700 system (Keyence, Osaka, Japan). Quantification was performed by a researcher who was blinded to the assay condition.

Cultured podocytes were transfected with a lentivirus expressing the full-length or active form of *Oasis* or venus (as a control)[17]. Two–four days after transfection, the podocytes were stained with Alexa Fluor™ 546-phalloidin (1:200) and DAPI (1:200). Quantification of phalloidin-staining patterns in the podocytes was performed by a researcher who was blinded to the experimental groups, based on a previous report[47]. Briefly, the intensity of phalloidin staining was grouped into four groups and scored: type A, more than 90% of the cell area filled with thick fibers; type B, more than 60% of the cell area filled with fine fibers; type C, shredded fibers; type D, no fibers in the cell area and flattened cell membrane.

**Lentivirus production**. Lentivirus expressing the N-terminal fragment of *Oasis/Creb3l1* was produced as previously described[17]. Lentivirus particles were produced by co-transfection with pNHP, pVSV-G, pCEP4-tat, and pCSII-EF containing the Full or N-terminal fragment of *OASIS/Creb3l1*, or control into 293 T cells using polyethylenimine (Polyethylenimine "Max", Polysciences, FL, USA). Cell culture media were changed to DMEM containing GlutaMAX™ (Thermo Fisher Scientific, Waltham, MA, USA) with serum at 9–14 h after the transfection. After 36 h, culture supernatant containing lentivirus was filtered through a 0.45 μm filter. The supernatant was centrifuged at 23,000 rpm for 2 h and the pellet was suspended with PBS. To induce overexpression of OASIS in podocytes, the cells were incubated in RPMI-1640 medium containing lentivirus vector, at a multiplicity of infection of 200–500, with 8 μg/mL polybrene (Nacalai Tesque) and 10% FBS, for 24 h. A lentivirus expressing *venus* was used as control.

**Albumin measurement**. Spot urine was obtained from the LPS-treated mice. In the DN model, twenty four-hour urine samples were collected using the metabolic cage system. Urinary albumin levels were measured using the LBIS Mouse Albumin ELISA Kit (FUJIFILM Wako Shibayagi, Gunma, Japan).

**Creatinine measurement**. Serum and urine creatinine were measured using the LabAssay™ Creatinine Kit (FUJIFILM Wako Pure Chemical Corporation, Osaka, Japan).

**Blood glucose measurement**. Blood samples were obtained from the tail, at 0, 4, and 8 weeks after STZ treatment. Blood glucose was measured using the Glucose CII Test Wako (FUJIFILM Wako Pure Chemical Corporation).

**Kidney histology**. Paraffin-embedded kidney sections were stained with PAS or Masson's trichrome. More than 100 tubules in five fields were randomly selected in each PAS-stained section, to evaluate tubular injury, according to a previous report[48]. The percentage of fibrosis per total tissue area was quantified in the Masson's trichrome-stained tissue sections, using ImageJ. Quantification was carried out by a researcher who was blinded to the experimental groups.

**Electron microscopy**. Kidney samples were fixed in 2.5% glutaraldehyde, and washed three times with cacodylate buffer, for 30 min. Sample preparation and photographing were consigned to Applied Medical Research Laboratory (Osaka, Japan). Ten podocyte foot processes in each mouse were randomly selected to evaluate foot process width by a researcher who was blinded to the experimental groups.

**Microarray**. Murine cultured podocytes were transfected with a lentivirus expressing the active form of *Oasis/Creb3l1* or *venus* for 24 h. Post the lentiviral infection, the medium was changed to RPMI-1640 with 10% FBS. Forty-eight

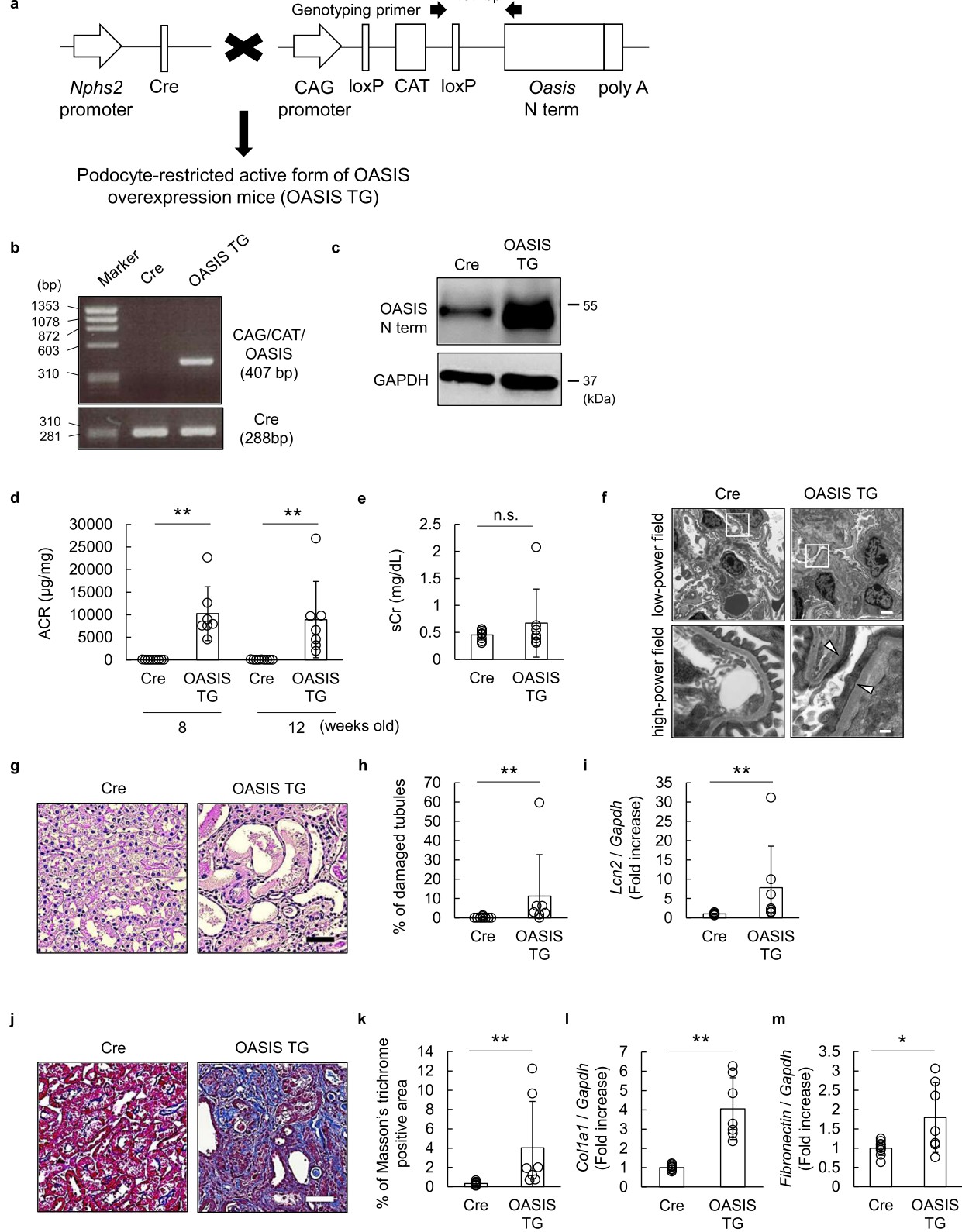

hours after treatment with the lentiviral vectors, total RNA was extracted from the OASIS-overexpressing podocytes, using the RNeasy® mini kit (Qiagen, Venlo, Netherlands). Gene expression was analyzed using the SurePrint G3 Mouse GE v2 8 × 60 K Microarray (Agilent Technologies, Santa Clara, CA, USA). The threshold was set to >2.0 or <0.5 times, and q < 0.1, to extract candidate downstream genes of OASIS, whose proteins could be secreted into the extracellular spaces.

**Quantitative PCR analysis**. Total RNA was prepared from kidneys or cells, using QIAzol® Lysis Reagent (Qiagen). Complementary DNA was synthesized from 1 μg total RNA using oligo(dT) primers (Thermo Fisher Scientific) and ReverTra Ace (Toyobo, Osaka, Japan). Gene expression was quantified using the Fast SYBR™ Green Kit (Applied Biosystems/Thermo Fisher Scientific). The primers used in this study were synthesized by Invitrogen/Thermo Fisher Scientific and are listed in Supplementary Table 1.

**Fig. 6 Kidney injury was induced in mice overexpressing the podocyte-restricted active form of OASIS. a** Schematic for the generation of mice overexpressing the podocyte-restricted active form of OASIS (OASIS TG). *Nphs2*-Cre mice (Cre) were used as control. **b** Representative genotyping results for Cre and OASIS TG mice. **c** Glomeruli were isolated from the kidneys of 12-week-old control and TG mice. Immunoblotting was performed using anti-OASIS and anti-GAPDH antibodies. Representative images are shown. **d, e** ACR (**d**) and sCr (**e**, 12-week-old) levels were measured in Cre or OASIS TG mice. Data are shown as mean ± SD ($n = 9$ for Cre and $n = 7$ for OASIS TG), *$P < 0.05$ and **$P < 0.01$, as analyzed using Student's *t*-test at each time-point. **f** Representative transmission electron microscopy images are shown at low and high magnifications. The arrowheads indicate podocyte foot process effacement. Scale bar: 2 μm (low-power field) and 200 nm (high-power field). **g** Representative images of Periodic acid–Schiff-stained kidneys of Cre and OASIS TG mice. Scale bar: 50 μm. **h** Quantitative assessment of tubular injury. Data are shown as mean ± SD ($n = 9$ for Cre and $n = 7$ for OASIS TG), **$P < 0.01$, as analyzed using Mann–Whitney *U* test. **i** The transcript expression of *Lcn2* was examined using quantitative PCR. The expression of the transcripts was normalized to that of *Gapdh*. Data are shown as mean ± SD ($n = 9$ for Cre and $n = 7$ for OASIS TG), **$P < 0.01$, as analyzed using Mann–Whitney *U* test. **j** Masson's trichrome staining was performed. Representative images are shown. Scale bar: 50 μm. **k** Fibrotic areas were measured. Data are shown as mean ± SD ($n = 9$ for Cre and $n = 7$ for OASIS TG), **$P < 0.01$, as analyzed using Mann–Whitney *U* test. **l, m** The transcript expression of *Col1a1* and *Fibronectin* was examined using quantitative PCR. The expression of the transcripts was normalized to that of *Gapdh*. Data are shown as mean ± SD ($n = 9$ for Cre and $n = 7$ for OASIS TG), *$P < 0.05$ and **$P < 0.01$, as analyzed using Student's *t*-test.

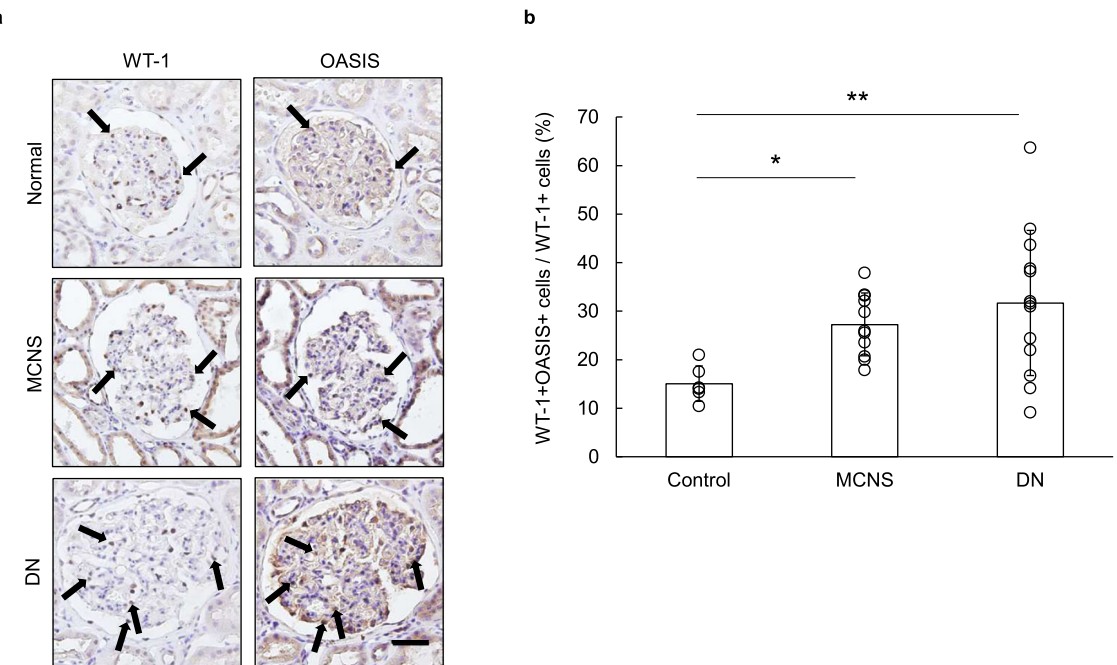

**Fig. 7 OASIS-positive podocytes were increased in the glomeruli of patients with human minimal change nephrotic syndrome and diabetic nephropathy. a** Immunohistochemical staining of kidneys from patients with MCNS and DN was performed using an anti-WT-1 or anti-OASIS antibody. Nuclei were stained with hematoxylin. Representative images are shown. Arrows: WT-1- and OASIS-positive cells. Scale bar: 100 μm. **b** The rate of OASIS/WT-1-double-positive cells to WT-1-positive cells in the glomeruli was measured. Data are shown as mean ± SD ($n = 6$ for normal, $n = 11$ for MCNS, and $n = 13$ for DN), *$P < 0.05$ and **$P < 0.01$, as analyzed using Dunnett test.

**Human kidney tissues**. Kidney biopsies from human subjects with MCNS ($n = 11$) and DN ($n = 13$), as well as human control ($n = 3$) samples were obtained from Chiba University Hospital. In addition, three control samples were purchased from OriGene Technologies Inc. (sample numbers CS701403, CS705934, and CS714584, Rockville, MD, USA). The study was conducted with informed consent and was approved by the Ethics Committee on Human Research of the Chiba University Hospital [approval no. 1119 (856)]. With respect to the purchased samples, the analysis was approved by the Clinical Research Ethics Review Committee of the Graduate School of Pharmaceutical Science, Osaka University [approval no. Yakuso30-4]. OASIS- or WT-1-positive cells were assessed using immunohistochemistry, as described above. The ratio of OASIS$^+$WT1$^+$ cells to WT1$^+$ cells was quantified, in a blinded fashion. Glomeruli with sclerosis were excluded from the measurement.

**Statistics and reproducibility**. Data are shown as mean ± SD of at least three independent experiments. For comparisons between two groups, Student's *t*-test or Mann–Whitney *U* test was used. For multiple comparisons, one-way analysis of variance followed by Dunnett test was used. Survival was estimated using the Kaplan–Meier method and log-rank test. $P < 0.05$ was considered to achieve statistical significance. Sample size and numbers are indicated in each figure legend.

**Reporting summary**. Further information on research design is available in the Nature Research Reporting Summary linked to this article.

## Data availability
The data that support the findings of this study are available from the corresponding author, M.O., upon reasonable request. Uncropped and unedited blot/gel images are provided in Supplementary Fig. 9. Microarray data have been deposited to the Gene Expression Omnibus (GEO) and are available at the accession number GSE206525. Source data underlying plots shown in figures are provided in Supplementary Data 1.

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

## Acknowledgements

We thank Kazuma Higashisaka, Atsushi Kasai, Yu-Shi Tian, Takafumi Nakae, and Takeo Harada (Osaka University) for their excellent technical assistance. We also thank Reiko Kizaki (Osaka University) for her excellent administrative work. This study was supported by Center for Medical Research and Education, Graduate School of Medicine, Osaka University. This study was partially supported by MEXT/JSPS KAKENHI grants 18K15027 to M.O., and 18H02603 & 20K21575 to Y.F. This study was also partially supported by grants from the Takeda Science Foundation to M.O., Japanese Association of Dialysis Physicians to M.O., and Platform Project for Supporting Drug Discovery and Life Science Research [Basis for Supporting Innovative Drug Discovery and Life Science Research (BINDS)] from AMED (grant numbers JP21am0101084, JP21am0101123, 22ama121052, and 22ama121054).

## Author contributions

Designing research studies: M.O. and Y.F. Conducting experiments: Y.M., M.O., A.Y., S.N., K.T., H.S., N.T., C.M., and G.S. Acquiring data: Y.M., M.O., A.Y., S.N., K.T., and S.T. Analyzing data: Y.M., M.O., A.Y., S.T., M.M., Y.O., and Y.F. Providing reagents: S.K., K.I., and K.A. Writing the manuscript: Y.M., M.O., A.Y., and Y.F.

## Competing interests

The authors declare no competing interests.

**Additional information**

