## [Peer Review File · Communications Biology]

Reviewers' comments:

Reviewer #1 (Remarks to the Author):

In their very well designed comprehensive study the Authors present novel findings on the role of the OASIS/CREB3L1 transcription factor in podocytes. To my knowledge, this is the first published study that provides details on the effects of OASIS expression on podocyte functions in vitro and in vivo. Moreover, the Authors show that mechanisms of OASIS-associated signal transduction involve modulation of PKC ζ that mediates interaction between podocytes and tubules. Presented in the manuscript results clearly indicate that expressed in podocytes OASIS contributes to kidney disorders. Finally, observations made in the experimental models were confirmed in the glomeruli obtained from the kidneys of the MCNS and DN patients. The results are well discussed.

However, after the presence, activation and involvement of podocytic OASIS in renal impairment has been shown, many new questions arise, such as the physiological role of OASIS in podocytes, possible association of OASIS with upregulated TGF beta in diabetic podocytopathy, and many other. Therefore I am sure that the present study will be followed by numerous other experiments.

All data provided are very well analyzed, using adequate statistical methods.

The graphs and photos are of a very good quality, clear and easy to interpret.

Minor concerns

1. Experimental methods are well presented, except for the immunofluorescence analysis (Suppl Fig 4). The Authors should provide some data how the IF intensity was measured (e.g. software)
2. Supplementary Fig 4 legend: It is not clear what is type A, type B, type C, type D
3. The Authors showed that in mice, overexpression of active OASIS led to albuminuria, whereas in cultured OASIS-overexpressing podocytes the actin cytoskeleton was disrupted.
4. The TGF beta – induced OASIS cleavage/ activation mediates some of the effects of TGF beta. On the other hand, exposure of podocytes to TGF beta has been shown to decrease podocin and nephrin expression and to induce proteinuria. So, it seems likely that in OASIS- overexpressing mice increased permeability to albumins might be caused by the decreased expression of the podocyte slit diaphragm proteins. Therefore, the IF staining of podocin and nephrin in cultured podocytes couldParticularly because the Authors have already tested the role of OASIS on Nphs1 and Nphs2 expression.
5. Discussion, line 188: “Finally, OASIS-positive podocytes were increased in kidneys of MCNS and DN patients.” In OASIS expressing cells, the factor plays its certain roles under physiological conditions. As far as I know, in podocytes these roles have not been fully elucidated yet , however I suppose that all healthy podocytes should be OASIS-positive . I guess that under pathological conditions it is OASIS expression but not number of OASIS-positive podocytes which increases. Therefore, I also wonder why in Figure 7B it is only ca. 15% Control podocytes that are OASIS-positive. Or, if OASIS is expressed in response to injury only, why the Control cells expressed all that much OASIS?

6. Few misspellings and errors, such as Page 10 line 168 “was occurred”; Page 10 line 170 “sever”

Barbara Lewko

Reviewer #2 (Remarks to the Author):

In this manuscript, the authors test the role of podocyte-specific OASIS in inducing tubular injury. They provide supporting data using constitutive podocyte-specific OASIS^{-/-} mice and OASIS transgenic mice. While there is some interesting data provided in this manuscript, there are several major concerns that significantly dampen the overall enthusiasm for this proposal.

Major Comments:

- 1) A major weakness is the lack of mechanistic studies that demonstrate the podocyte-tubular crosstalk in the setting of modulating OASIS expression in podocytes.
- 2) Changes in serum Cr post-LPS treatment are modest, and therefore the biological significance of the data is significantly dampened (Fig 2, 4G).
- 3) The use of LPS model is not a reliable model to demonstrate the extent of kidney injury, as c57bl/6 mice are fairly resistant to kidney injury in this model.
- 4) The rigor of experimental data to support tubular injury is very minimal- immunostaining and quantification of markers such as lotus lectin, aquaporin 1 is necessary to support the authors conclusions regarding the extent of tubular injury.
- 5) Validation of podocyte-specific OASIS knockout is not strong. Immunostaining with podocyte-specific marker or western blot of isolated podocytes is necessary to confirm podocyte-specific knockout of OASIS, especially since OASIS is expressed in other cell types in the kidney.
- 6) In Fig. 1G- confirmation that OASIS induction post-LPS treatment is specific to podocytes is unclear. Validation with immunostaining and quantification is necessary.
- 7) It is unclear whether tubular expression of OASIS is altered in podocyte-specific OASIS^{-/-} with and without LPS treatment.
- 8) While urine PRKCI is increased in urine of OASIS ko mice, is it also upregulated in podocytes in OASIS ko mice?

9) HK-2 cells are not a reliable model of tubular cells, primary tubular cells should be used to support the data provided.

10) It remains unclear why in Fig. 6 there is no change in serum Cr if the proposed model is podocyte-tubular crosstalk. In addition, there is a change in Lcn2, suggesting tubular injury, but no serum Cr?

11) The human biopsy data is relatively weak with small sample sizes. In addition, it appears that OASIS expression is induced in all cell types, suggesting the lack of specificity of the antibody used for staining in human biopsies.

12) Line 170, states “data not shown” – which is not appropriate.

13) There are several spelling mistakes in the manuscript than significantly detracted from the review.

Our point-by-point responses to the comments are described below.

The responses to the comments

Reviewer #1:

Minor concerns

Comment 1: Experimental methods are well presented, except for the immunofluorescence analysis (Suppl Fig 4). The Authors should provide some data how the IF intensity was measured (e.g. software)

Response: We thank the reviewer. We are sorry that we forgot to mention this detail. We have now added the information on the immunofluorescence methods to the revised manuscript.

Comment 2: Supplementary Fig 4 legend: It is not clear what is type A, type B, type C, type D

Response: We apologize for the confusing presentation. We have now added the method for quantification of staining patterns to the revised manuscript.

Comment 3: The Authors showed that in mice, overexpression of active OASIS led to albuminuria, whereas in cultured OASIS-overexpressing podocytes the actin cytoskeleton was disrupted.

Response: We agree with you. Podocyte foot process effacement, which is closely associated with cytoskeletal abnormalities, was observed in the podocyte-restricted OASIS TG mice. Upon examination of the expression of genes related to the cytoskeleton and slit diaphragm, including *nephrin*, *Cd2ap*, *Kirrel*, and *Actn4*, we observed no difference in these expression levels between the whole kidneys of OASIS TG and control mice. We are currently in the process of identifying other factors downstream of OASIS that might be involved in podocyte dysfunction, and will report this as a separate study. We have now removed the data on gene expression of slit diaphragm-related factors in OASIS-overexpressing podocytes (Supplementary Table 2 of the original manuscript) from the revised manuscript, to avoid confusion.

There was no difference in the expression of genes related to the cytoskeleton and slit diaphragm between the OASIS TG and Cre (control) mice.

Comment 4: The TGF beta – induced OASIS cleavage/ activation mediates some of the effects of TGF beta. On the other hand, exposure of podocytes to TGF beta has been shown to decrease podocin and nephrin expression and to induce proteinuria. So, it seems likely that in OASIS- overexpressing mice increased permeability to albumins might be caused by the decreased expression of the podocyte slit diaphragm proteins. Therefore, the IF staining of podocin and nephrin in cultured podocytes could provide some additional information about the mechanisms of OASIS-dependent effects. Particularly, because the Authors have already tested the role of OASIS on Nphs1 and Nphs2 expression.

Response: We apologize for the confusing explanation. We have previously reported that TGF- β 1 induces OASIS expression in myofibroblasts [*FASEB J.* 2021;35(2):e21158]. In this study, we demonstrated that OASIS was induced upon LPS treatment in podocytes. As pointed out by the reviewer, we stimulated cultured podocytes with TGF- β 1, which did not upregulate OASIS expression in podocytes, as described below. Therefore, we consider that OASIS signaling network is differentially regulated, in a cell type-dependent manner.

According to the reviewer’s comment, we examined the expression of podocin and nephrin in OASIS-overexpressing podocytes, as described below. The expression levels did not change in OASIS-overexpressing podocytes, as compared to those in the control group, suggesting that OASIS may not regulate the expression of podocin and nephrin. We are currently in the process of identifying factors downstream of OASIS that might be involved in podocyte dysfunction, and will report the same in the form of a separate study.

TGF- β 1 did not upregulate OASIS expression in podocytes.

Murine cultured podocytes were stimulated with TGF- β 1 and/or LPS for 6 or 24 h. Immunoblotting was performed using anti-OASIS and anti-GAPDH antibodies. Representative images are shown.

OASIS overexpression did not affect the expression of nephrin and podocin in cultured podocytes.

Murine cultured podocytes were transfected with a lentivirus expressing the active form of *Oasis*, or *venus* (as a control). Immunofluorescence analysis was performed using an anti-nephrin or anti-podocin antibody (red), and DAPI (blue).

Comment 5: Discussion, line 188: “Finally, OASIS-positive podocytes were increased in kidneys of MCNS and DN patients.” In OASIS expressing cells, the factor plays its certain roles under physiological conditions. As far as I know, in podocytes these roles have not been fully elucidated yet, however I suppose that all healthy podocytes should be OASIS-positive. I guess that under pathological conditions it is OASIS expression but not number of OASIS-positive podocytes which increases. Therefore, I also wonder why in Figure 7B it is only ca. 15% Control podocytes that are OASIS-positive. Or, if OASIS is expressed in response to injury only, why the Control cells expressed all that much OASIS?

Response: We agree with your concerns about the frequency of OASIS-positive cells.

We think this might be due to 3 possibilities:

First, as stated in the text, in this study, we assessed podocyte OASIS expression in consecutive sections, without staining a single section with DAPI and antibodies against WT-1 and OASIS together. Of note, OASIS is localized in the nuclei. Thus, in this study, we simply evaluated the ratio of the number of OASIS-positive nuclei in one section to the number of WT-1-positive podocytes in the consecutive section, but not the ratio of the number of OASIS-positive nuclei to the number of WT-1-positive podocyte nuclei in the same section.

Second, immunohistochemical analysis has revealed that OASIS was stained in podocyte nuclei of *in vivo* kidney sections. Though it is well known that OASIS is activated and translocated into the nucleus to function, the mechanism for clearance of OASIS from nuclei is unknown. It may be possible that only about 15% of the activated form of OASIS is detected at steady state.

Another explanation is that the frequency of OASIS-positive podocytes may not have been as high, because damage in the glomeruli, especially under the condition of diabetic nephropathy, does not occur uniformly. At least, the number of podocytes with detectable OASIS increased in the kidneys of MCNS and DN patients as well as in those of LPS-treated mice, as highlighted by the data in the revised manuscript. We have now revised the corresponding text in the *Discussion* section of the revised manuscript.

With respect to the physiological function, podocyte-restricted OASIS-deficient mice did not show any obvious abnormality during the experimental period. Therefore, the physiological function of OASIS remains unclear. We are planning to examine the long-term effects of OASIS expression under physiological conditions, by analyzing the aging model of OASIS cKO mice.

Comment 6: Few misspellings and errors, such as Page 10 line 168 “was occurred”; Page 10 line 170 “sever”

Response: We apologize for the mistakes. We had now taken the help of an English proofreading service to edit the English text. Attached below is a proofreading certificate for the manuscript.

Reviewer #2:

Comment 1: A major weakness is the lack of mechanistic studies that demonstrate the podocyte-tubular crosstalk in the setting of modulating OASIS expression in podocytes.

Response: We appreciate your valuable suggestion. To confirm that OASIS exhibits detrimental effects on tubular cells, mechanistically through PRKCI, we examined the effects of the PRKCI inhibitor on OASIS cKO mice. OASIS cKO mice were treated with the PRKCI inhibitor, post LPS injection. Interestingly, the tubular-protective effect in OASIS cKO was significantly abolished by the PRKCI inhibitor, suggesting that OASIS knockout in podocytes reduced the LPS-induced tubular injury, at least by increasing PRKCI. We have now added the data concerning this to Figure 5I–K of the revised manuscript.

Comment 2: Changes in serum Cr post-LPS treatment are modest, and therefore the biological significance of the data is significantly dampened (Fig 2, 4G).

Response: Thank you for your comment. Serum creatinine level did not increase dramatically after LPS (10 mg/kg) treatment, consistent with previous reports [*Kidney Int.* 2012;82(1):53-9. *Theranostics.* 2019;9(2):405-423]. This is an experimental limitation. Indeed, we also used higher doses of LPS (12.5 and 15 mg/kg) to induce severe kidney injury, but all the mice died 24 h after such LPS treatment. In the future, we need to explore other renal disease models associated with podocyte OASIS, in order to further clarify the pathophysiological significance of OASIS.

Comment 3: The use of LPS model is not a reliable model to demonstrate the extent of kidney injury, as c57bl/6 mice are fairly resistant to kidney injury in this model.

Response: We agree with you. As known, cumulative evidence indicates that mice of the C57Bl/6 background are significantly more resistant to kidney injury than Balb/c mice, in several kidney disease models [*Nephrology (Carlton).* 2011;16(1):30-8]. We believe that this suggestion is very important, and hence we have addressed this in the *Discussion* section. Further studies are needed to investigate the effects of podocyte OASIS using genetically modified mice of the Balb/c background. We are currently working on backcrossing OASIS cKO mice with Balb/c mice, and will present the results for the same as a future study. With respect to the comments 2 and 3 of the reviewer, we have now mentioned that the roles of podocyte OASIS in various kidney diseases will be clarified in future studies, and that OASIS is expected to become a potential therapeutic target for kidney diseases.

Comment 4: The rigor of experimental data to support tubular injury is very minimal- immunostaining and quantification of markers such as lotus lectin, aquaporin 1 in necessary to support the authors conclusions regarding the extent of tubular injury.

Response: Thank you for your helpful comment. As suggested by the reviewer, we stained kidney sections with FITC-conjugated lotus tetragonolobus lectin (LTL), to evaluate the extent of tubular injury. No difference was observed in the LTL-positive area between OASIS cKO mice and control mice, upon LPS treatment. In the first place, tubular detachment did not occur and the expression pattern of LTL was not changed after LPS treatment. Secondly, the LTL-positive area increased significantly in the kidneys of OASIS cKO mice, as compared to those of control mice, in the diabetic nephropathy model. We have now added this data to the revised manuscript.

Comment 5: Validation of podocyte-specific OASIS knockout is not strong. Immunostaining with podocyte-specific marker or western blot of isolated podocytes is necessary to confirm podocyte-specific knockout of OASIS, especially since OASIS is expressed in other cell types in the kidney.

Response: We completely agree with you. Since the properties of podocytes grown in culture were altered compared to those of the original podocytes [*Kidney Int.* 2006;69(11):2101-6], the protein expression of OASIS in the glomeruli was evaluated immediately after isolation from the OASIS cKO mice and control mice, by means of western blot with an anti-OASIS monoclonal antibody. The protein expression of OASIS was significantly decreased in the glomeruli of OASIS cKO mice, suggesting that OASIS expression was at least reduced in the podocytes of OASIS cKO mice. We have now added this data to Figure 1D of the revised manuscript. Additionally, *Oasis* mRNA expression was quantitatively measured using real-time PCR. *Oasis* mRNA was reduced to less than 10% in the glomeruli of OASIS cKO mice, as compared to those of control mice, as described in Figure 1C of the revised manuscript. As you know, *Neph-Cre* mice are commonly used for podocyte-specific genetic modification (*Genesis*. 2003;35:39-42., *J Am Soc Nephrol.* 2017;28:2654-2669). These data indicated that in glomeruli, OASIS is expressed mainly in podocytes.

Comment 6: In Fig. 1G- confirmation that OASIS induction post-LPS treatment is specific to podocytes is unclear. Validation with immunostaining and quantification is necessary.

Response: We agree with your comment. As stated in the text, this study suffers from a limitation of not being able to carry out fluorescence multiple staining using the OASIS antibody. Therefore, it was difficult to accurately evaluate the intensity of the OASIS staining. The number of OASIS-positive podocytes was evaluated and found to increase in the kidneys of LPS-treated mice, as compared to those of control mice, using serial sections. LPS upregulates OASIS expression, at least in podocytes *in vivo*, although it may also increase OASIS expression in other cells of the glomeruli. We have added this data to Figure 1I and 1J of the revised manuscript, in addition to describing the same in the text.

Comment 7: It is unclear whether tubular expression of OASIS is altered in podocyte-specific OASIS^{-/-} with and without LPS treatment.

Response: Thank you for your comment. As suggested by the reviewer, tubular epithelial cells were isolated from OASIS cKO and control mice, with or without LPS treatment. Immunoblotting revealed that OASIS expression was not significantly changed in the tubular cells of cKO and control mice, with or without LPS treatment, as described below. We will examine the mechanisms of induction of OASIS expression and the role of OASIS in tubular cells in future studies.

OASIS expression was not significantly altered in the tubular cells of cKO and control mice, with or without LPS treatment.

Tubular epithelial cells were isolated from the kidneys of LPS-treated OASIS cKO and control mice. Immunoblotting was performed using an anti-OASIS antibody. The experiment was repeated twice.

Comment 8: While urine PRKCI is increased in urine of OASIS ko mice, is it also upregulated in podocytes in OASIS ko mice?

Response: Thank you for your valuable comment. As suggested by the reviewer, we evaluate the expression of PRKCI and WT-1, a podocyte marker, in the kidneys, by means of immunostaining using serial sections. PRKCI expression was observed in the glomeruli, including in podocytes and the tubulointerstitial space. The intensity of PRKCI staining in the glomeruli tended to increase in the OASIS cKO mice, though it was not statistically significant. Since our results showed that PRKCI is secreted in the urine, we believe that no clear difference in PRKCI expression was observed in the kidney tissues. We have now added this data to Supplementary Figure 4.

Comment 9: HK-2 cells are not a reliable model of tubular cells, primary tubular cells should be used to support the data provided.

Response: We sincerely appreciate your helpful comment. As pointed out by the reviewer, tubular cells were isolated from murine kidneys and treated with LPS, in the presence or absence of PRKCI. Similar results were obtained in primary murine tubular cells, as seen in HK-2 cells. We have now replaced the data obtained using HK-2 cells with the data obtained using isolated murine tubular cells in the revised manuscript (Figure 5H).

Comment 10: It remains unclear why in Fig. 6 there is no change in serum Cr if the proposed model is podocyte-tubular crosstalk. In addition, there is a change in *Lcn2*, suggesting tubular injury, but no serum Cr?

Response: Thank you for your precious comment. Interestingly, podocyte-restricted overexpression of the active form of OASIS resulted in lower survival rate and body weight, than those seen in the control mice, at 12 weeks of age. Therefore, by the age of 8 weeks, mice with albumin creatinine ratio exceeding 15,000 $\mu\text{g}/\text{mg}$ were defined as the severe group, and their renal function was evaluated. It was found that the serum creatinine level was elevated in the severe group. Compared to OASIS Tg mice that survived to 12 weeks of age, the degree of increase in *Lcn2* mRNA expression was greater in the severe group, relative to that in the control group. These data indicated that the

severity of tubular/glomerular injury in the Tg mice may be linked to the decrease in their renal function. We have now added these data to Supplementary Figure 6 and 7 of the revised manuscript.

Comment 11: The human biopsy data is relatively weak with small sample sizes. In addition, it appears that OASIS expression is induced in all cell types, suggesting the lack of specificity of the antibody used for staining in human biopsies.

Response: We agree with you. As suggested by the reviewer, we repeated the experiments with as many samples as we could. Based on our previous report [*FASEB J.* 2021;35(2):e21158] and the Human Protein Atlas database, OASIS is also expressed in the tubular epithelial cells of human kidneys. We confirmed that OASIS was not detected in the tubular epithelial cells, when immunohistochemistry was performed without the use of the OASIS antibody. We believe that the expression of OASIS in the tubular epithelial cells may be involved in the progression of kidney injury. We are now investigating the roles of OASIS in injured tubular cells using cell-specific OASIS knockout mice, and will discuss this in future papers. We have now described this in the *Discussion* section of the revised manuscript.

Comment 12: Line 170, states “data not shown” – which is not appropriate.

Response: We completely agree with you, and have now shown the same data in Supplementary Figure 8.

Comment 13: There are several spelling mistakes in the manuscript than significantly detracted from the review.

Response: We apologize for the mistakes. We have now taken the help of an English proofreading service to edit the English text. Attached below is a proofreading certificate for the manuscript.

Editing Certificate

This document certifies that the manuscript listed below has been edited to ensure language and grammar accuracy and is error free in these aspects. The logical presentation of ideas and the structure of the paper were also checked during the editing process. The edit was performed by professional editors at Editage, a division of Cactus Communications. The sections titled references were not edited by Editage upon the author's request.

The author's core research ideas were not altered during the editing process. Editage guarantees the quality of editing with the assumption that our suggested changes have been accepted and the edited text has not been altered without the knowledge of our editors.

MANUSCRIPT TITLE

Upregulation of OASIS/CREB3L1 in podocytes contributes to the disturbance of kidney homeostasis

AUTHORS

Yoshiaki Miyake; Masanori Obana; Ayaha Yamamoto; Shunsuke Noda; Koki Tanaka; Hibiki Sakai; Narihito Tatsumoto; Chihiro Makino; Soshi Kanemoto; Go Shioi; Shota Tanaka; Makiko Maeda, Yoshiaki Okada; Kazunori Imaizumi; Katsuhiko Asanuma, Yasushi Fujio

ISSUED ON

May 02, 2022

JOB CODE

KCTGT_2

Vikas Narang

Vikas Narang
Chief Operating Officer - Editage

Editage, a brand of Cactus Communications, offers professional English language editing and publication support services to authors engaged in over 1300 areas of research. Through its community of experienced editors, which includes doctors, engineers, published scientists, and researchers with peer review experience, Editage has successfully helped authors get published in internationally reputed journals. Authors who work with Editage are guaranteed excellent language quality and timely delivery.

GLOBAL :

+1(833) 979-0061 | request@editage.com

JAPAN :

0120-50-2987 | submissions@editage.com

REVIEWERS' COMMENTS:

Reviewer #2 (Remarks to the Author):

The Authors have very carefully addressed all my questions and comments. In order to clarify some results , additional Western blot and immunostaining procedures have been performed and results have been discussed. The manuscript has been completed and changed according to my suggestions. Also, language proofreading improved the quality of the manuscript. Therefore, my suggestion is to accept the manuscript in the revised form.

Reviewer #3 (Remarks to the Author):

The Reviewer's concerns were addressed very accurately and the Authors supplemented the manuscript with additional explanation. In response to the Reviewer's concerns, the Authors also completed their results by performing additional in vivo and in vitro experiments and laboratory tests and analyses. Additional results have been included in the revised manuscript. Corresponding figures have been modified and additional comments have been added in the text. According to the Reviewer's suggestions, professional English proofreading has been completed. Taken together, in my opinion the current version of revised manuscript is suitable for publication

Manuscript #: COMMSBIO-21-2978A

Our point-by-point responses to the comments are described below.

Reviewer #2:

Comment: The Authors have very carefully addressed all my questions and comments. In order to clarify some results, additional Western blot and immunostaining procedures have been performed and results have been discussed. The manuscript has been completed and changed according to my suggestions. Also, language proofreading improved the quality of the manuscript. Therefore, my suggestion is to accept the manuscript in the revised form.

Response: We sincerely appreciate your time and careful peer review.

Reviewer #3:

Comment: The Reviewer's concerns were addressed very accurately and the Authors supplemented the manuscript with additional explanation.

In response to the Reviewer's concerns, the Authors also completed their results by performing additional in vivo and in vitro experiments and laboratory tests and analyses. Additional results have been included in the revised manuscript. Corresponding figures have been modified and additional comments have been added in the text.

According to the Reviewer's suggestions, professional English proofreading has been completed.

Taken together, in my opinion the current version of revised manuscript is suitable for publication

Response: We sincerely appreciate your time and careful peer review.